# Bacterial Profile and Antibiotic Resistance among Cancer Patients with Urinary Tract Infection in a National Tertiary Cancer Hospital of Nepal

**DOI:** 10.3390/tropicalmed6020049

**Published:** 2021-04-13

**Authors:** Gambhir Shrestha, Xiaolin Wei, Katrina Hann, Kyaw Thu Soe, Srinath Satyanarayana, Bhola Siwakoti, Shankar Bastakoti, Rashmi Mulmi, Kritika Rana, Nirmal Lamichhane

**Affiliations:** 1Department of Community Medicine, Maharajgunj Medical Campus, Institute of Medicine, Tribhuvan University, Kathmandu 44600, Nepal; 2Dalla Lana School of Public Health, University of Toronto, Toronto, ON M5T 3M7, Canada; xiaolin.wei@utoronto.ca; 3Sustainable Health Systems, Freetown 42215, Sierra Leone; hann.katrina@gmail.com; 4Department of Medical Research (Pyin Oo Lwin Branch), Ministry of Health and Sports, Nay Pyi Taw 15011, Myanmar; kyawthusoe.dmr@gmail.com; 5Center for Operational Research, International Union against Tuberculosis and Lung Disease, 75000 Paris, France; ssrinath@theunion.org; 6Department of Cancer Prevention, Control and Research, B.P. Koirala Memorial Cancer Hospital, Chitwan 44200, Nepal; siwakotibhola@gmail.com; 7Department of Pathology, B.P. Koirala Memorial Cancer Hospital, Chitwan 44200, Nepal; drshankarbastakoti@gmail.com; 8World Health Organization, Kathmandu 44600, Nepal; mulmir@who.int (R.M.); ranak@who.int (K.R.); 9Department of Surgical Oncology, B.P. Koirala Memorial Cancer Hospital, Chitwan 44200, Nepal; nirmal.lamichhane@gmail.com

**Keywords:** neoplasms, urinary tract infections, antimicrobial resistance surveillance, oncology, operational research, SORT-IT, AWaRe

## Abstract

Cancer patients are at high risk of antibiotic resistant bacterial urinary tract infections (UTIs). In this study, we assessed the bacterial profile and antibiotic resistance among cancer patients suspected of UTI in B.P. Koirala Memorial Cancer Hospital in Nepal through a cross-sectional study with routinely collected data. All cancer patients who had a recorded urine culture between July 2018–June 2019 were included in the study. Out of 308 patients who had undergone culture, 73 (24%) of samples had bacterial growth. The most common organisms isolated were *E. coli* (58%), *Staphylococcus* (11%) and *Klebsiella* (10%). These bacteria had undergone susceptibility testing to 27 different antibiotics in various proportions. Of the limited antibiotic testing levels, nitrofurantoin (54/66, 82%) and amikacin (30/51, 59%) were the most common. Among those tested, there were high levels of resistance to antibiotics in the “Access” and “Watch” groups of antibiotics (2019 WHO classification). In the “Reserve” group, both antibiotics showed resistance (polymyxin 15%, tigecycline 8%). Multidrug resistance was seen among 89% of the positive culture samples. This calls for urgent measures to optimize the use of antibiotics in UTI care at policy and health facility levels through stewardship to prevent further augmentation of antibiotic resistance among cancer patients.

## 1. Introduction

There is increasing evidence from different parts of the world of the high prevalence of antimicrobial resistance (AMR) in cancer patients [1,2,3,4,5]. Patients with cancers are immunosuppressed and therefore at a high risk of serious opportunistic infections [6]. Cancer treatments, including chemotherapy, surgery and radiation, put patients at significantly elevated risk of opportunistic infections and subsequent infection-related death [7,8]. Furthermore, high consumption of antibiotics and prolonged hospital stays make cancer patients vulnerable to multidrug-resistant (MDR) bacteria strains. Furthermore, without effective use of antibiotics for prevention and treatment of infections, the success of major surgeries and cancer chemotherapies would be compromised, putting them at even higher risk. Although rational use of antibiotics has been proven effective in reducing AMR [9,10], frequent and irrational use of antibiotics has been linked to rising MDR bacteria among cancer patients [11].

Cancer, with an estimated incidence of 103.7 per 100,000 population in 2018, is a major public health problem in Nepal [12]. At least 700,000 deaths globally have been attributed to AMR each year, which is predicted to rise to 10 million deaths each year by 2050 [13]. The burden of cancer in Nepal has been increasing over the last decade [14] with the top cancers being lung, cervical, breast, stomach and colorectal [15]. On the other hand, the AMR burden is also increasing in Nepal due to weak regulations and rampant irrational use of antibiotics in health facilities and over-the-counter purchase of antibiotics in communities [16,17]. Risk of AMR in cancer patients requires special attention of hospital management, clinicians and public policy makers as it escalates the difficulty in treatment, the costs of treatment and poor prognosis.

Urinary tract infection (UTI) is one of the most common infections among cancer patients due to their prolonged immunosuppression, complex cancer treatment and catherization [18,19]. Treating UTI in cancer patients is clinically challenging as many cancer patients are at high risk of AMR due to long-term chemotherapies, depressed immune systems and repeated use of antibiotics to prevent and/or treat infections [20]. Therefore, understanding patterns of AMR resistance for UTI is imperative to inform clinical practice and stewardship in relation to the appropriate use of antibiotics in cancer patients. Such evidence is essential to guide the development of clinical guidelines, as well as to provide recommendations for policies to build up the AMR stewardship program in support of optimization of the Use of Antibiotics in Nepal’s National AMR Containment Action Plan [21].

To the best of our knowledge, there is no study in Nepal assessing the burden and pattern of AMR in cancer patients. Therefore, in this study, we investigated the pattern of bacterial isolates and AMR among cancer patients with suspected UTI. Our specific objectives were: (a) to describe the demographic and clinical factors of all cancer patients who had undergone urine culture in B.P Koirala Memorial Cancer Hospital between July 2018 to June 2019; (b) to determine the proportion of urine samples with bacterial growth, describe the bacterial species and assess the patient factors associated with urine culture positive results in those patients; and (c) to describe antibiotics to which the bacteria were tested for resistance in vitro and their resistance levels.

## 2. Materials and Methods

### 2.1. Study Design

This was a cross-sectional study, with secondary analysis of routinely collected hospital data.

### 2.2. Setting

#### 2.2.1. General Setting

Nepal is a landlocked country situated in Southeast Asia with an estimated population of 29 million in 2020. Cancer care is available in both the public and private sectors; however, there are few hospitals dedicated to providing specialized oncology services. The Ministry of Health and Population (MoHP) provides NRS 100,000 (USD 1000) of free services for cancer patients and patients are required to pay out-of-pocket once the allocated budget is surpassed. Nepal has recently started its journey on the path to an integrated response to address the challenges of AMR. The government has developed the National AMR Containment Action Plan in 2016.

#### 2.2.2. Study Setting

This study was conducted in B.P. Koirala Memorial Cancer Hospital (BPKMCH), Bharatpur, Chitwan, Nepal. It is a National Cancer Hospital, located at the center of Nepal with a capacity of 450 beds. Approximately 125,000 patients visited the outpatient department in 2017, out of which 5172 were cancer patients. The majority of cancer patients in Nepal are treated in this hospital. The most common presentations among cancer patients are lung, cervical and breast cancers [22]. BPKMCH keeps the records of diagnosed cancer patients in its record section by providing a unique patient ID, which helps to distinguish them from non-cancer patients.

### 2.3. Antibiotics Susceptibility Testing

The antibiotic susceptibility pattern of the isolates is determined by the Kirby–Bauer disk diffusion method according to Clinical and Laboratory Standards Institute (CLSI) guidelines [23]. The results are recorded as susceptible (S), intermediate (I) and resistant (R). In this study, intermediate results were merged into the susceptible category. The antibiotics purchased from HiMedia (India) were used for drug susceptibility test. The reference strain used as quality control was *E. coli* (ATCC 25922) and *S. aureus* (ATCC 25923).

### 2.4. Study Population and Period

The study population was all cancer patients who had their urine culture done during the study period of 1 July 2018 to 30 June 2019 in BPKMCH. If a patient had multiple urine culture reports during the study period, only the first culture results were included in the analysis. This allowed for assessment of the initial AMR pattern of cancer patients in their first culture completed in the BPKMCH and to avoid duplication.

### 2.5. Variables, Data Collection and Entry

Data were collected from the microbiology laboratory registers of the cancer hospital. We identified patients who had a culture report at the microbiology unit and those were linked to their electronic medical records using bill number. This aided the distinction between registered cancer patients from others. From the medical records, we listed the cancer patient’s ID number and outpatient department number. From this information we accessed the patient’s paper-based file and extracted demographic and clinical variables. If we were not able to find the patient’s file, then we (1) cross checked with the medical record section if the patient was currently admitted; and (2) followed up at the medical section once a month over a period of two months. In case of discrepancy in demographic variables between matched medical records and laboratory reports, we used data from the medical record.

Variables included demographic and clinical characteristics of patients such as age, gender, type of cancer, stage of cancer, duration since diagnosis, type of cancer treatment, presence of fever and antibiotics used at the time of urine sample collection. If the urine culture results showed bacterial growth, then the name of the bacterial species, the antibiotics to which drug susceptibility testing was done and its results were noted. Cancer was classified into: (a) solid tumor, defined as tumors with an abnormal mass of tissue comprising of sarcomas and carcinomas; and, (b) hematological tumor, defined as cancer that begins in blood-forming tissue, such as the bone marrow, or in the cells of the immune system. This study used the 2019 WHO AWaRe Classification of classifying antibiotics (used for assessing the bacterial drug susceptibility) which consists of three stewardship groups: “Access”, “Watch” and “Reserve” [24]. “Access” group antibiotics include antibiotics that have activity against a wide range of commonly encountered susceptible pathogens and should be available at all health facilities. “Watch” group antibiotics should not be used unless “Access” antibiotics are not effective. Antibiotics in the “Reserve” group should be treated as “last resort” options, when all alternatives have failed or are not suitable. Multidrug resistant (MDR) infection was defined as resistance to three or more classes of antibiotics.

Data were collected using a structured form developed by the investigation team. Data were single entered on a structured proforma using EpiData entry software version 3.1 (EpiData Association, Odense, Denmark) by first author with the help of his assistant reading aloud from the form. A total of 30 randomly selected forms were checked for any error in the data entry.

### 2.6. Statistics Analysis

Data were analyzed using EpiData analysis version X (EpiData Association, Odense, Denmark) and Stata 15.0 (StataCorp LP, College Station, TX, USA). Data were summarized using the two descriptive statistics, frequency and percentage. To assess the association between demographic and clinical characteristics associated with culture-positive results, we used log binomial models with robust variance estimators. The association was expressed in culture-positive prevalence and adjusted culture-positive prevalence ratios. A p-value less than 0.05 was considered as statistically significant.

## 3. Results

### 3.1. Demographic and Clinical Profile of Cancer Patients

A total of 1271 urine cultures were performed during the study period, out of which 401 reports were of cancer patients. A total of 93 reports had to be excluded as they were re-test samples and would cause duplication in the study. Hence, 308 urine culture reports of cancer patients were included in this study. The demographic and clinical profiles of the cancer patients who were included in the study are given in Table 1. Nearly half (53%) were below the age of 45 years, most (62%) were males, more than half (58%) had solid tumors and the remaining (40%) had hematological tumors. Nearly half of them had been diagnosed with cancer for one to two years and the majority of them (45%) were on chemotherapy. At the time of urine sample collection, only 39% were recorded as having fever and 20% were on antibiotics. Data on fever (present or absent) and whether the patients were on antibiotics or not were unknown for 45% and 65% respectively. Information regarding cancer types is provided in the Appendix A.

### 3.2. Proportion of Bacterial Growth and Their Profile

Of the 308 patients whose urine was subjected to culture, bacterial growth was observed in 73 (24%) samples and their profile is given in Table 2. The most common organisms isolated were *E. coli* (58%), *Staphylococcus* (11%) and *Klebsiella* (10%).

### 3.3. Factors Associated with Urine Culture Positivity

The demographic and clinical factors associated with bacterial growth/isolation are also given in Table 1. On bivariate analysis, age group > 45 years (when compared to those < 15 years), those with solid tumors (when compared to hematological cancers), those who had undergone surgery (when compared to those who underwent chemotherapy) had higher prevalence of bacterial growth/isolation. However, on multivariable analysis, none of these factors were associated with bacterial growth/isolation.

### 3.4. Antibiotic Resistance Level

Table 3 provides details on the antibiotic drug susceptibility of the bacteria isolated from the urine samples. Of the 73 culture-positive samples, only two samples were sensitive to all the antibiotics being tested. Culture-positive samples from 65 (89%) patients indicated MDR urine infection, with 22 having resistance to three classes of antibiotics, 19 resistant to four classes, 14 resistant to five classes, 5 resistant to six classes and 5 resistant to seven and more classes of antibiotics. The bacteria were subjected to drug susceptibility testing on 27 different antibiotics in various proportions. There were only seven antibiotics to which at least 50% of the bacterial growth/isolates underwent drug susceptibility testing. These antibiotics were nitrofurantoin, ampicillin, amikacin, sulfamethoxazole/trimethoprim and gentamycin from the “Access” group and ciprofloxacin and ceftriaxone from the “Watch” group. Out of these seven antibiotics, the resistance levels among isolates were found to be the lowest in nitrofurantoin (18%) and amikacin (41%). For the other five antibiotics, the resistance levels were very high, ranging from 55% to 90%. Resistance to fluoroquinolones was high among the limited number of isolates tested for resistance, for example, 83% for ciprofloxacin, 69% for ofloxacin, 100% for norfloxacin and 73% for levofloxacin. High resistance levels were also seen among the cephalosporin drugs. Of note, 13 and 12 isolates were tested for polymyxin B and tigecycline, the two antibiotics from the “Reserve” group, reporting resistance rates of 15% and 8% respectively.

The antibiotic testing and resistance pattern disaggregated by the different types of bacteria is given in Appendix A.

## 4. Discussion

Our study is the first of its kind to describe bacterial growth and antibiotic resistance patterns in cancer patients with UTI in Nepal. Out of the 308 patients with urine culture results, one fourth had bacterial growth, most commonly *E. coli*. Of the 73 samples having bacterial isolates, we observed high levels of resistance to ampicillin, amoxicillin and cefalexin of antibiotics in the “Access” group (>80%) and to fluroquinolones in the “Watch” group (>63%). In addition, we observed resistance in the “Reserve” group, on average in 11.5%. We also observed MDR among 89% of the culture positive samples.

A number of quality improvement points were noticed in this study. First, we could not determine whether all cancer patients had undergone systematic screening for UTI or which sub-group of patients underwent urine culture and drug susceptibility testing in our setting. Therefore, based on the information that 308 cancer patients have undergone urine culture in the one-year period in this hospital, we are unable to comment on prevalence of UTI in cancer patients in our setting. Furthermore, we are also unable to comment on whether the 24% bacterial growth seen in our patient population samples is high or low. Previous studies from other countries have shown that growth of bacteria among urine samples from cancer patients suspected of UTI ranged from 6% in a hospital in Ethiopia to as high as 72% in a hospital in Egypt [25,26]. Within Nepal, the bacterial growth in urine samples from general patients ranges from 14% to 32% in those who have undergone urine bacterial cultures [27,28].

Second, a large proportion of data were missing on the three most important variables that were previously known to predict bacterial growth in urine specimens, namely presence or absence of symptoms, stage of cancer and use of antibiotics prior to the time of requesting urine culture [4]. Possible reasons may be that there is no standardized proforma for requesting urine cultures and, hence, clinicians may not have observed any guidelines in requesting urine cultures. The fact that we did not find any correlation between patient characteristics and bacterial growth in urine cultures may be due to the large proportion of missing data. Previous studies have shown that old age (when compared to younger age), patients with solid tumors (when compared to hematological tumors) and patients who have undergone surgical treatment (when compared to non-surgical treatment) were at higher risk of UTI [4,29], findings similar to those of this study.

Third, we did not identify any pattern in antibiotic susceptibility testing of these bacterial species. To our knowledge, microbiologists are not routinely given indications by clinicians on which drugs they prefer for antibiotic drug susceptibility testing. In this situation the microbiologists choose the most available antibiotics testing discs to test resistance. Frequent shortages of drug susceptibility tests further exacerbates the situation. All these issues explain the lack of a standard pattern in antibiotic susceptibility testing and have been previously documented to interfere in antibiotic susceptibility testing [30].

Lastly, due to the irregular pattern of antibiotic testing, the prevalence of resistance to various antibiotics could not be ascertained in this study.

Our results indicated a high resistance level among bacteria isolates among the cancer patients in Nepal. Isolates of our samples were resistant to most antibiotics in the “Access” group such as ampicillin, amoxicillin, sulfamethoxazole/trimethoprim and cefalexin. Of a more important note, there are high resistance rates to fluroquinolones and third generation cephalosporins in the “Watch” group and some were resistant to antibiotics in the “Reserve” group. This finding was in line with other studies [19,20,31]. This points to the need for optimal and rational use of antibiotics in cancer patients to prevent antibiotic resistance, as well as improvement of quality of antibiotic resistance testing. Currently, no guidance exists on symptoms indicating urine sample culture in cancer patients and antibiotics which require drug susceptibility testing in the cancer hospital.

This study is the first of its kind in Nepal presenting the burden and pattern of AMR in cancer patients. The major strength of this study is that we have used data from routine clinical/programmatic conditions using standard data extraction practices. Therefore, we believe the study findings to be reflective of ground-level reality. There are some limitations of the study. First, this study was conducted in only one tertiary care hospital in the country. Apart from this hospital, there are several other health facilities that offer cancer care to people in Nepal and we are unable to generalize the study findings beyond this health facility. Second, due to resource constraints, we could not include a qualitative component to this study to better understand and explain the study findings. Hence, most of the explanations for the study findings are anecdotal.

The implications of the study findings for policy and practice to improve UTI care for cancer patients are: First, we recommend prospective research studies to ascertain the prevalence of UTI, current antibiotic use/prescription patterns for UTI and antibiotic resistance patterns among cancer patients in a representative sample of health facilities that provide cancer care in the country. Second, we recommend the implementation of standard protocols for systematic screening of cancer patients for UTI, standardized proforma for requesting urine cultures containing the relevant clinical details of the patients, systematic testing of bacteria for antibiotic drug susceptibility testing, recording and periodic reporting of drug resistance patterns and rational use of antibiotics in cancer patients. There is urgent need for an AMR stewardship program to educate and create awareness among health care professionals and the community on the rationale use of antibiotics [32].

## 5. Conclusions

Our study in the largest cancer hospital in Nepal indicated a high level of antibiotic resistance among cancer patients with possible UTI. We also identified gaps such as the need to improve urine culture and antibiotic drug susceptibility testing in the cancer hospital. The finding of drug resistance to many antibiotics of the “Watch” and “Reserve” group of WHO classification is of serious concern of AMR in cancer patients. This calls for urgent measures to optimize UTI care and antibiotic administration in health facilities and prevent further augmentation of antibiotic resistance among cancer patients.

## Figures and Tables

**Table 1 tropicalmed-06-00049-t001:** Demographic and clinical factors and their association with bacterial growth/isolation in cancer patients who had undergone urine culture in B.P Koirala Memorial Cancer Hospital, Bharatpur, Nepal between July 2018 to June 2019 (n = 308).

Demographic and Clinical Characteristics	Total		Culture-Positive		Culture-Positive Prevalence Ratio (95% CI)	Adjusted Culture Positive Prevalence Ratio (95% CI)	*p*-Value
	N	(%) *	n	(%) ^#^			
**Total**	**308**	**(100)**	**73**	**(24)**			
**Age**							
<15 years	71	(23)	8	(11)	Ref	Ref	
15–29 years	52	(17)	12	(23)	2.04 (0.90–4.65)	1.32 (0.50–3.45)	0.57
30–44 years	40	(13)	10	(25)	2.21 (0.95–5.17)	1.14 (0.40–3.23)	0.79
45–59 years	64	(21)	16	(25)	2.21 (1.02–4.83)	1.18 (0.43–3.22)	0.73
≥60 years	81	(26)	27	(33)	2.95 (1.43–6.09)	1.43 (0.54–3.79)	0.46
**Gender**							
Male	192	(62)	43	(22)	Ref		
Female	116	(38)	30	(26)	1.15 (0.77–1.73)	1.30 (0.77–2.18)	0.32
**Classification of Cancer**							
Hematological	123	(40)	14	(11)	Ref	Ref	
Solid	178	(58)	57	(32)	2.81 (1.64–4.82)	2.24 (0.69–7.20)	0.17
Not available	7	(2)	2	(29)	2.51 (0.70–8.96)	Not estimated	
**Duration diagnosis**							
<1 year	98	(32)	21	(21)	Ref	Ref	
1–2 years	161	(52)	41	(25)	1.18 (0.74–1.88)	1.37 (0.77–2.43)	0.27
>2 years	49	(16)	11	(22)	1.04 (0.54–1.99)	1.18 (0.50–2.78)	0.70
**Cancer stage**							
Stage I	6	(2)	3	(50)	2.19 (0.95–5.04)	3.66 (0.98–13.65)	0.05
Stage II	14	(5)	5	(36)	1.56 (0.74–3.28)	1.19 (0.45–3.16)	0.72
Stage III	8	(3)	1	(13)	0.54 (0.08–3.49)	0.36 (0.04–2.86)	0.33
Stage IV	21	(7)	5	(24)	1.04 (0.47–2.32)	1.11 (0.41–2.99)	0.82
Not recorded	259	(84)	59	(23)	Ref	Ref	
**Type of cancer treatment**							
Only chemotherapy	139	(45)	18	(13)	Ref	Ref	
Only surgery	43	(14)	18	(42)	3.23 (1.85–5.64)	1.88 (0.60–5.81)	0.27
Only radiation	6	(2)	2	(33)	2.57 (0.76–8.65)	1.92 (0.34–10.67)	0.45
Surgery + chemotherapy	62	(20)	18	(29)	2.24 (1.25–4.01)	1.23 (0.41–3.65)	0.70
Chemotherapy + radiation	14	(5)	4	(29)	2.20 (0.86–5.62)	1.36 (0.34–5.37)	0.65
Surgery + radiation	13	(4)	3	(23)	1.78 (0.60–5.26)	0.86 (0.18–4.10)	0.85
Surgery + chemotherapy + radiation	15	(5)	8	(53)	4.11 (2.17–7.82)	2.33 (0.66–8.20)	0.18
Not recorded/not initiated on Rx	16	(5)	2	(13)	0.96 (0.24–3.79)	8.11 (not estimated)	0.98
**Febrile at the time of urine collection**							
No	48	(16)	8	(17)	Ref	Ref	
Yes	121	(39)	31	(26)	1.53 (0.76–3.10)	2.62 (0.95–7.25)	0.06
Not recorded	139	(45)	34	(24)	1.46 (0.73–2.95)	2.49 (0.84–7.37)	0.09
**Antibiotics prescribed prior test for suspected UTI**							
Yes	61	(20)	19	(31)	Ref	Ref	
No	50	(16)	9	(18)	0.57 (0.28–1.16)	0.70 (0.29–1.67)	0.43
Not recorded	197	(64)	45	(23)	0.73 (0.46–1.15)	0.66 (0.29–1.52)	0.33

* Column percentage, ^#^ Row percentage. CI: confidence interval; Rx: treatment; UTI: urinary tract infection.

**Table 2 tropicalmed-06-00049-t002:** Bacteria isolated from urine culture samples of cancer patients in B.P Koirala Memorial Cancer Hospital, Bharatpur, Nepal between July 2018 to June 2019 (n = 73).

Bacteria Isolated	Total	
	n	(%)
*E. coli*	42	(58)
*Staphylococcus*	8	(11)
*Klebsiella*	7	(10)
*Enterococci*	6	(8)
*Citrobacter*	6	(8)
*Pseudomonas*	3	(4)
*Proteus*	1	(1)

**Table 3 tropicalmed-06-00049-t003:** Results of antibiotic resistance testing on bacteria (n = 73) isolated from urine samples of cancer patients in B.P Koirala Memorial Cancer Hospital, Bharatpur, Nepal between July 2018 to June 2019.

Antibiotic	Isolates Tested		Resistance among Isolates Tested	
	n	(%)	n	(%)
**Access group**				
Nitrofurantoin	66	(90)	12	(18)
Ampicillin	52	(71)	47	(90)
Amikacin	51	(70)	21	(41)
Sulfamethoxazole/trimethoprim	46	(63)	36	(78)
Gentamycin	40	(55)	22	(55)
Cefalexin	33	(45)	31	(94)
Amoxicillin/clavulanic acid	32	(44)	27	(84)
Ampicillin/sulbactam	3	(4)	1	(33)
Doxycycline	1	(1)	0	(0)
**Watch group**				
Nalidixic acid	8	(11)	5	(63)
Ciprofloxacin	53	(73)	44	(83)
Norfloxacin	15	(21)	15	(100)
Ofloxacin	32	(44)	22	(69)
Levofloxacin	15	(21)	11	(73)
Ceftriaxone	40	(55)	31	(78)
Cefixime	35	(48)	34	(97)
Cefotaxime	16	(22)	15	(94)
Cefepime	32	(44)	26	(81)
Cefoperazone	3	(4)	2	(67)
Cefuroxime	10	(14)	10	(100)
Ceftazidime	5	(7)	4	(80)
Meropenem	1	(1)	1	(100)
Imipenem	9	(12)	1	(11)
Vancomycin	10	(14)	6	(60)
Piperacillin/tazobactam	11	(15)	9	(82)
**Reserve group**				
Polymyxin B	13	(18)	2	(15)
Tigecycline	12	(16)	1	(8)

## Data Availability

Data can be made available on considerable request to corresponding author.

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
