# Peer review of "Bacterial Profile and Antibiotic Resistance among Cancer Patients with Urinary Tract Infection in a National Tertiary Cancer Hospital of Nepal"

_tropicalmed, 2021, doi:10.3390/tropicalmed6020049_

Round 1

Reviewer 1 Report

Authors wrote an very interesting paper from Nepal on important issue antimicrbial resistance and antibiotic use.

I think that the topic is crucial also in a major global health approuch and the setting of study is very difficult. I suggest to accept the paper after minor revision

Below my suggestions:

  1. Introduction: do you have any data about global burdenof AMR in your Country? Introduce better the setting of study and why cancer patietns are most fragile. Furthermore research on AMR in low setting is crucial to save vulnerable people (cancer patietns, children with undernutrition, women under caesarean section.
  2.  Methods and results: are very clear and also statystical analisys is good and with quality
  3. Discussion : add some consideration about education on antimicrobial resistance in doctor and expecailly youg doctor that will be the prescriber of antibioticts for the future 40 ys. Their role is very crucial and education on this topic is very central to fight antimicrobial reisstance in your contry and in other part of world (see and cite Italian young doctors' knowledge, attitudes and practices on antibiotic use and resistance: A national cross-sectional survey. J Glob Antimicrob Resist. 2020 Dec;23:167-173. doi: 10.1016/j.jgar.2020.08.022.)

Furthermore propose one publich health action that came from your very interesting and well wrote paper

Author Response

Ref: Manuscript ID: tropicalmed-1151137

Dear Editor,

I, on behalf of all the authors, would like to thank you and the reviewers for your comments/suggestions for making this manuscript a better one. We have tried to address all the comments (highlighted in red in the revised manuscript) to our fullest ability. In the enclosed response letter, we have provided our point-to-point reply to the reviewers’ comments. We hope that our replies and revisions meet your expectation. If there are any further suggestions/comments, we will be happy to incorporate them in our manuscript.

Gambhir Shrestha

gamvir.stha@gmail.com

Author’s responses to reviewers’ comments/suggestions

Reviewer 1

  1. Reviewer’s comment: Authors wrote an very interesting paper from Nepal on important issue antimicrbial resistance and antibiotic use. I think that the topic is crucial also in a major global health approuch and the setting of study is very difficult. I suggest to accept the paper after minor revision.

Authors’ reply: Thank you very much.

  1. Reviewer’s comments: Introduction: do you have any data about global burden of AMR in your Country? Introduce better the setting of study and why cancer patietns are most fragile.

Authors’ reply: We have added the global and Nepal’s burden of AMR (revised manuscript line numbers 65-66 and 68-71).

Line 52-59 describes the setting of the study and why cancer patients are most fragile for AMR “Patients with cancers are immunosuppressed, and therefore at a high risk of serious opportunistic infections [6]. Cancer treatments, including chemotherapy, surgery, and radiation, put patients at significantly elevated risk of opportunistic infections and subsequent infection-related death [7,8]. Furthermore, high consumption of antibiotics and prolonged hospital stays make cancer patients vulnerable to multidrug-resistant (MDR) bacteria strains. Also, without effective use of antibiotics for prevention and treatment of infections, the success of major surgeries and cancer chemotherapies would be compromised, putting them at evener high-risk.”

  1. Reviewer’s comment: Furthermore research on AMR in low setting is crucial to save vulnerable people (cancer patietns, children with undernutrition, women under caesarean section.

Authors’ reply: We strongly agree: research and their implications can save vulnerable people from AMR specially in low-income setting.

  1. Reviewer’s comment: Methods and results: are very clear and also statystical analisys is good and with quality

Authors’ reply: Thank you.

  1. Reviewer’s comment: Discussion: add some consideration about education on antimicrobial resistance in doctor and expecailly youg doctor that will be the prescriber of antibioticts for the future 40 ys. Their role is very crucial and education on this topic is very central to fight antimicrobial reisstance in your contry and in other part of world (see and cite Italian young doctors' knowledge, attitudes and practices on antibiotic use and resistance: A national cross-sectional survey. J Glob Antimicrob Resist. 2020 Dec;23:167-173. doi: 10.1016/j.jgar.2020.08.022.)

Authors’ reply: Thank you for bringing out this issue. We have added your suggestion with the reference (revised manuscript line numbers: 338-340).

  1. Reviewer’s comment: Furthermore propose one public health action that came from your very interesting and well wrote paper.

Authors’ reply: From this operational research we have proposed further prospective research studies to ascertain the prevalence of UTI, current antibiotic use/prescription patterns for UTI and antibiotic resistance pattern among cancer patients in a representative sample of health facilities that provide cancer care in the country. Second, we recommend the implementation of standard protocols for systematic screening of cancer patients for UTI, standardized proforma for requesting urine cultures containing the relevant clinical details of the patients, systematic testing of bacteria for antibiotic drug susceptibility testing, recording and periodic reporting of drug resistance patterns and rational use of antibiotics in cancer patients (revised manuscript line numbers: 330-337).

Reviewer 2 Report

This study is a cross-sectional review of patients (a majority are men), most of them under 60 years, with a slight majority of solid tumours over haematological cancer, and a majority receiving chemotherapy, a group of them in combination with solid tumour surgery. This means a heterogeneous population. Moreover, there is an important weakness in terms of tumour stage reporting, which further complicate the interpretation in terms of severity of the individual cancer disease and risk factors.

As mentioned by the authors in the discussion (a too long text on this issue – could be shortened), this study gives neither epidemiological data such as incidence and prevalence of UTI in cancer patients in Nepal, nor underlying causes (e.g. catheterisation at surgery, gender, impact of immunosuppression, co-morbidity.)

Therefore, this study can be analysed only in terms of results of urine culture and related sensitivity pattern. This is however an important descriptive report showing a resistance pattern over 75% for most important antimicrobial agents such as ciprofloxacin, cephalosporines, piperacillin/tazo; >60% for Vancomycin, etc… This leaves most patients with complex infections in conjunction with cancer diagnosis and on-going treatment without any active antibiotics.

Another weakness is the lake of reported out-come of the treatment of UTI (febrile or not).

The article could benefit from a focus on the possible causes in Nepal for this important resistance pattern, the implications on infectious disease treatment in general and cancer disease patients – at high risk of complicated infections - management in particular, and on higher risk of deadly out-come (e.g. sepsis,. Febrile UTI, pneumonia), and how an antimicrobial stewardship program could through education and awareness on a national level could lead to a more rational use of antibiotics.

Another secondary, co-lateral, observation is the weakness in cancer stage reporting and the importance of stringent case reporting in the national system. For good research this is imperative.

Author Response

Ref: Manuscript ID: tropicalmed-1151137

Dear Editor,

I, on behalf of all the authors, would like to thank you and the reviewers for your comments/suggestions for making this manuscript a better one. We have tried to address all the comments (highlighted in red in the revised manuscript) to our fullest ability. In the enclosed response letter, we have provided our point-to-point reply to the reviewers’ comments. We hope that our replies and revisions meet your expectation. If there are any further suggestions/comments, we will be happy to incorporate them in our manuscript.

Gambhir Shrestha

gamvir.stha@gmail.com

Author’s responses to reviewers’ comments/suggestions

Reviewer 2

  1. Reviewer’s comment: There is an important weakness in terms of tumour stage reporting, which further complicate the interpretation in terms of severity of the individual cancer disease and risk factors.

Authors’ reply: Thank you. We agree that the recording of tumour stage was inadequate. We have highlighted this important finding in our manuscript (revised manuscript line numbers 284-287) and have recommended actions (to standardize recording and reporting formats) to improve the same (revised manuscript line numbers 333-337).

  1. Reviewer’s comment: As mentioned by the authors in the discussion (a too long text on this issue – could be shortened), this study gives neither epidemiological data such as incidence and prevalence of UTI in cancer patients in Nepal, nor underlying causes (e.g. catheterisation at surgery, gender, impact of immunosuppression, co-morbidity.)

Authors’ reply: We agree with your comment that our study does not provide information on incidence, prevalence and risk factors for UTI among cancer patients. Nevertheless, our study is an initial step to understand the epidemiology of UTI among cancer patients as it is a poorly explored/researched area in our country setting. As this study was done among the patients with urine culture, it provides information on the prevalence of AMR/culture positivity among the tested. Luckily, gender has been captured in our study. We understand that many risk factors were missing, and we have acknowledged this aspect in our manuscript (revised manuscript line numbers 284-291). We strongly feel that our study will provide valuable guidance for the design of more robust research studies in the future. We hope the reviewer agrees to our point of view on this aspect.

  1. Reviewer’s comment: Therefore, this study can be analysed only in terms of results of urine culture and related sensitivity pattern. This is however an important descriptive report showing a resistance pattern over 75% for most important antimicrobial agents such as ciprofloxacin, cephalosporines, piperacillin/tazo; >60% for Vancomycin, etc… This leaves most patients with complex infections in conjunction with cancer diagnosis and on-going treatment without any active antibiotics.

Authors’ reply: We fully agree upon this.

  1. Reviewer’s comment: Another weakness is the lake of reported out-come of the treatment of UTI (febrile or not).

Authors’ reply: The status of fever at the time of urine culture is an important predictor for our study, however it was found to be under reported/poorly documented. We have mentioned this aspect in our study (revised manuscript line numbers 284-287)

  1. Reviewer’s comment: The article could benefit from a focus on the possible causes in Nepal for this important resistance pattern, the implications on infectious disease treatment in general and cancer disease patients – at high risk of complicated infections - management in particular, and on higher risk of deadly out-come (e.g. sepsis, Febrile UTI, pneumonia), and how an antimicrobial stewardship program could through education and awareness on a national level could lead to a more rational use of antibiotics.

Authors’ reply: Yes, we agree. We have incorporated the suggestions in the discussion part (revised manuscript line numbers 338-340).

  1. Reviewer’s comment: Another secondary, co-lateral, observation is the weakness in cancer stage reporting and the importance of stringent case reporting in the national system. For good research this is imperative.

Authors’ reply: Thank you for nicely bringing up this issue. Our operational research is an attempt at bringing this issue to the notice of the policy makers, hospital managers, clinicians and the laboratory.